New insights into the lifestyle of Allosaurus (Dinosauria: Theropoda) based on another specimen with multiple pathologies

Foth Christian 1 2 3 christian.foth@gmx.net
Evers Serjoscha W. 2 4
Pabst Ben 5
Mateus Octávio 6 7
Flisch Alexander 8
Patthey Mike 9
Rauhut Oliver W.M. 1 2
1 SNBS, Bayerische Staatssammlung für Paläontologie und Geologie , München , Germany
2 Department of Earth and Environmental Sciences, Ludwig-Maximilians-Universität , München , Germany
3 Department of Geosciences, University of Fribourg/Freiburg , Fribourg , Switzerland
4 Department of Earth Sciences, University of Oxford , Oxford , UK
5 Sauriermuseum Aathal , Aathal-Seegräben , Switzerland
6 CICEGe, Faculdade de Ciências e Tecnologia, FCT, Universidade Nova de Lisboa , Caparica , Portugal
7 Museu da Lourinhã , Rua João Luis de Moura, Lourinhã , Portugal
8 Swiss Federal Laboratories for Materials Science and Technology , Center for X-ray Analytics, Düebendorf , Switzerland
9 Vetsuisse Fakulty, Universität Zürich , Zürich , Switzerland
Wedel Mathew
Electronic publication date: 2015 May 12
Publication date: 2015
Volume: 3
Electronic Location ID: e940
Received 2015 Feb 4; Accepted 2015 Apr 16
Copyright: © 2015 Foth et al.
Copyright year: 2015
Copyright holder: Foth et al.
License: This is an open access article distributed under the terms of the Creative Commons Attribution License, which permits unrestricted use, distribution, reproduction and adaptation in any medium and for any purpose provided that it is properly attributed. For attribution, the original author(s), title, publication source (PeerJ) and either DOI or URL of the article must be cited.
License URL: https://creativecommons.org/licenses/by/4.0/

Keywords: Paleopathology, Gregarious behavior, Pseudarthrosis, Jurassic, Osteomyelitis, Archosauria, Theropoda

Funding: Volkswagen Foundation I/84 640 This study was supported by the Volkswagen Foundation under grant I/84 640 (to Oliver W.M. Rauhut). The funders had no role in study design, data collection and analysis, decision to publish, or preparation of the manuscript.

==============================
Adult large-bodied theropods are often found with numerous pathologies. A large, almost complete, probably adult Allosaurus specimen from the Howe Stephens Quarry, Morrison Formation (Late Kimmeridgian–Early Tithonian), Wyoming, exhibits multiple pathologies. Pathologic bones include the left dentary, two cervical vertebrae, one cervical and several dorsal ribs, the left scapula, the left humerus, the right ischium, and two left pedal phalanges. These pathologies can be classified as follows: the fifth cervical vertebra, the scapula, several ribs and the ischium are probably traumatic, and a callus on the shaft of the left pedal phalanx II-2 is probably traumatic-infectious. Traumatically fractured elements exposed to frequent movement (e.g., the scapula and the ribs) show a tendency to develop pseudarthroses instead of a callus. The pathologies in the lower jaw and a reduced extensor tubercle of the left pedal phalanx II-2 are most likely traumatic or developmental in origin. The pathologies on the fourth cervical are most likely developmental in origin or idiopathic, that on the left humerus could be traumatic, developmental, infectious or idiopathic, whereas the left pedal phalanx IV-1 is classified as idiopathic. With exception of the ischium, all as traumatic/traumatic-infectious classified pathologic elements show unambiguous evidences of healing, indicating that the respective pathologies did not cause the death of this individual. Alignment of the scapula and rib pathologies from the left side suggests that all may have been caused by a single traumatic event. The ischial fracture may have been fatal. The occurrence of multiple lesions interpreted as traumatic pathologies again underlines that large-bodied theropods experienced frequent injuries during life, indicating an active predatory lifestyle, and their survival perhaps supports a gregarious behavior for Allosaurus. Alternatively, the frequent survival of traumatic events could be also related to the presence of non-endothermic metabolic rates that allow survival based on sporadic food consumption or scavenging behavior. Signs of pathologies consistent with infections are scarce and locally restricted, indicating a successful prevention of the spread of pathogens, as it is the case in extant reptiles (including birds).

Introduction

Palaeopathology is the study of diseases and traumatic injuries in extinct animals and reveals great potential to provide insights into behavior (e.g., Rothschild & Storrs, 2003), physiology (e.g., Rothschild et al., 2003), life history (e.g., Hanna, 2002) as well as interspecific (e.g., predator–prey relationships) and intraspecific interactions (e.g., intraspecific combats or cannibalism) (e.g., Carpenter, 2000; Tanke & Currie, 2000; Currie, 2000; Rogers, Krause & Rogers, 2003; Avilla, Fernandes & Ramos, 2004; Carpenter et al., 2005; Farke, Wolff & Tanke, 2009; Butler et al., 2013; Hone & Tanke, 2015). In recent years, the study of osteological pathologies among non-avian dinosaurs has become of great interest, documenting a wide range of different kinds of injuries and diseases, e.g., fractures and stress fractures (Rothschild, 1988; Rothschild, Tanke & Ford, 2001; Hanna, 2002; Anné et al., 2014), amputations (Farke & O’Connor, 2007; Butler et al., 2013), bite marks and scratches (Carpenter, 2000; Tanke & Currie, 2000; Peterson et al., 2009; Bell, 2010), cancer and tumor growth (Rothschild et al., 2003; Arbour & Currie, 2011), developmental disorders (Witzmann et al., 2008) as well as different kinds of microbial infections (Hanna, 2002; Wolff et al., 2009; Witzmann et al., 2011). Of special interest in this respect are non-lethal pathologies, as they can potentially tell us something about the lifestyle of the animal. Especially, large-bodied non-avian theropods are frequently found with numerous fractures, bite marks and infections (Gilmore, 1920; Molnar & Farlow, 1990; Molnar, 2001; Hanna, 2002; Brochu, 2003; Farke & O’Connor, 2007; Rothschild & Molnar, 2008; Bell, 2010; Bell & Coria, 2013), indicating an active predatory life style predisposed to injuries (Hanna, 2002). The basal tetanuran Allosaurus is one of the best-documented dinosaurs in this field of research (e.g., Marsh, 1884; Gilmore, 1920; Moodie, 1923; Petersen, Isakson & Madsen, 1972; Madsen, 1976; Rothschild, 1988; Rothschild & Martin, 1993; Hanna, 2002; Anné et al., 2014). However, only one study, which is based on the almost complete Allosaurus specimen MOR 693 (‘Big Al’) as well as isolated material from the Cleveland-Lloyd Dinosaur Quarry, has studied its pathologies in greater detail and in a comparative approach (Hanna, 2002). Here, we report a second almost complete, probably adult Allosaurus specimen from the Upper Jurassic of Wyoming, U.S.A, which possesses several pathologic bones, including the left dentary, two mid-cervical vertebrae, a right cervical rib, several dorsal ribs, the left scapula, the left humerus, the right ischium, and the left pedal phalanges II-2 and IV-1 (Fig. 1). After documentation and diagnosis, the single pathologies of the specimen will be compared with the data from Hanna (2002) and that of other large-bodied theropods, so that the current study provides new insights into the disease patterns and lifestyles of these remarkable predators.

Figure 1 Overview of pathologies in SMA 0005.

Skeletal reconstruction of SMA 0005, showing all pathologic bones. Pathologic elements from the left side are shown in red, while respective elements from the right side are marked in blue. Unpaired pathologic bones are colored in cyan. Green ribs represent ribs from the left, for which a pathologic condition is uncertain. Abbreviations: c, cervical; cr, cervical rib; de, dentary; dr, dorsal rib; hu, humerus; is, ischium; p, pedal phalanx; sc, scapula. Skeletal reconstruction of SMA 0005 with courtesy from the Sauriermuseum Aathal.

Material and Methods

The Allosaurus specimen SMA 0005 (‘Big Al 2’) was collected from the Upper Jurassic outcrops of the Morrison Formation (Late Kimmeridgian—Early Tithonian) of the Howe Ranch (Howe Stephens Quarry), Big Horn County, Wyoming, by a team of the Sauriermuseum Aathal (Switzerland) in 1996, close to the famous Howe Quarry discovered by Barnum Brown in 1934 (Brown, 1935; Breithaupt, 1997). The almost complete skeleton was found partially articulated and probably represents an adult individual (total body length = 7.6 m), which is about 12% larger than MOR 693 (‘Big Al’), which was found only a few hundred meters away.

For classification of different pathologies present in SMA 0005 we follow the nomenclature of Hanna (2002), who classifies osteological abnormalities as (1) traumatic (resulting from traumatic injury), (2) infectious (resulting from viral, bacterial and protozoan infection), (3) traumatic-infectious (resulting from secondary infection of an injured element), (4) developmental (caused by growth disturbance during development), and (5) idiopathic (pertaining to a condition without clear pathogenesis).

Traumatic injuries of bone include fractures and amputations. If these injuries do not cause the immediate death of an animal they are characterized by healing responses, usually in form of callus formation (Cleas, Wolf & Augat, 2000), which is proliferating growth of originally non-mineralized connective tissue to close the gap and stabilize the respective injury (Park et al., 1998; Cleas, Wolf & Augat, 2000; Schell et al., 2005). Generally, the callus surrounds the perimeter of the injured bone locally and forms a different superficial structure compared to healthy bone. If the healing process of the injury is not disturbed by secondary infections or interfragmentary movements, the callus is remodelled by zonal lamellar bone after some time (McKibbin, 1978; Park et al., 1998). In case of bone fractures, however, intense mechanical loadings and interfragmentary movements can rupture the bridging callus tissue, including its vessels, resulting in the formation of a pseudarthrosis or ‘false joint’ (Cleas, Wolf & Augat, 2000; Loboa, Beaupré & Carter, 2001; Klein et al., 2003; Strube et al., 2008), which is usually accompanied by chronic pain, and often so by disability (Loboa, Beaupré & Carter, 2001). However, pseudarthrosis can also result from syn-traumatic malunions (Klein et al., 2003).

An osteological abnormality caused by viral, bacterial or protozoan infections is called osteitis. If such infection becomes chronic and affects the bone marrow it is called osteomyelitis (Pschyrembel, 1990), which is usually characterized by comb-like lesions on the bone surface. In extant mammals, tissue-invasive microbial infections are often characterized by locally restricted, subperiostal suppurative abscesses. In later stages, these abscesses can cause necroses of original bone due to an infiltration of pus into the blood vessel system, impairing the blood supply of the local bone area. Such infiltrations can further lead to a spread of microbial pathogens via the blood stream, affecting other skeletal elements (so called haematogenous osteomyelitis) (Ortner & Putschar, 1981; Pschyrembel, 1990; Gross, Rich & Vickers-Rich, 1993). In contrast, extant reptiles (including birds) do not respond to tissue-invasive microbial infections by producing liquid pus (Montali, 1988; Rega, 2012), but instead by exuding fibrin into the infected areas, which forms local fibriscesses (as a type of granuloma), and preventing the spread of the infection via the blood stream (Gomis et al., 1997; Huchzermeyer & Cooper, 2000; Cooper, 2005). Thus, reptiles usually manifest only contiguous osteomyelitis. Besides osteomyelitis, osteitis can also lead to the formation of exostoses, superficial bony outgrowths.

Developmental disorders are pathologies related to ontogenetic abnormalities resulting from inherent genetic defects or growth disturbances, whereas in idiopathic abnormalities the cause of the osteological pathology is unknown (Hanna, 2002).

To study potential internal structures several pathologic bones of SMA 0005 were CT scanned. The left dentary and the left scapula were investigated using a Siemens SOMATOM Sensation Open (CT) system at Vetsuisse Faculty (University of Zurich) with source: 120 kV, 176 mA, rotation time: 1 s, pitch: 0.55 mm and slice thickness: 0.6 mm. The fifth cervical was scanned with a 450 kV X-ray system MG450 (YXLON) and a CITA 101B+ industrial CT scanner with a collimated line detector (CITA Systems Inc., Pueblo, Colorado, USA) at the Center for X-ray Analytics (EMPA, Swiss Federal Laboratories for Materials Science and Technology) with source: 450 kV, 3.3 mA, focal spot size: 1.0 mm, target: wolfram, 750 projections of 0.04 s over 360°, slice thickness: 0.25 mm. The generated CT data were preceded with help of the 3D reconstruction software package Amira 5.3.3 (Visage Imaging, Inc., San Diego, California, USA) and Mimics 16.0 (Materialise HQ, Leuven, Belgium). Unfortunately, the foot was firmly installed in the mounted skeleton, so that the pathologic phalanges could not be scanned.

To allow readers their own assessments of the pathologic structures described, we created a supplementary information file, showing all figures with interpretive lines in a clear version (see Supplemental Information 1).

Results

Dentary

Description. The anterior end of the left dentary in SMA 0005 is strongly modified. In lateral view, it has the shape of an anterior rosette, similar to the morphology seen in spinosaurid megalosaurs (Stromer, 1915; Charig & Milner, 1997; Sereno et al., 1998), measuring c. 110 mm in anteroposterior length. The anterior part is both dorsally and ventrally expanded, reaching a maximal height at the level of the fifth teeth mounted (in both dentary most teeth are not original). Anteriorly, the alveolar border curves ventrally, forming a convex arch (Figs. 2A and 2B, see also S1 in the Supplemental Information 1). This morphology clearly differs from the normal condition in Allosaurus, where the anterior portion of the dental margin is slightly convex (Fig. 2C). As a consequence of the morphology in the anterior end of the left dentary, the symphysial region of the mandible is dorsoventrally shortened, and when both mandibles are aligned with the ventral border of the symphysis, at least the anterior part of the left alveolar margin would project dorsally well beyond the right alveolar margin. Ventral to the first two teeth mounted, a V-shaped depression is present on the medial side of the bone, which is c. 10 mm deep, 45 mm long and 30 mm high. At the level of the first tooth the depression curves anterodorsally, reaching the dental margin of the dentary (Fig. 2B). The opening at the margin measures c. 15 mm in dorsoventral length. Based on the location, this structure might represent a further tooth position. However, CT data show that the anterior part of the left dentary is formed by compact bone, while more posterior parts on the same dentary shows repetitive indentations representing deep alveoli (Fig. 3). Consequently, the presence of the first two teeth mounted in the left dentary cannot be verified by the CT data. On the other hand, it is also not possible to identify the nature of the medial depression with help of the CT data. No clear indication of a fracture, bite marks, callus or other lesion is visible.

Figure 2 The dentaries of SMA 0005.

(A) Left dentary with pathologic anterior end in lateral view. (B) Medial side of the left dentary with pathologic anterior end in mirrored view. (C) Right dentary in mirrored view, showing the normal condition for Allosaurus. The differences in the shape of the alveolar margin in both dentaries (A, C) are shown with a dashed line. Note that most teeth in both dentaries are not original, but glued to the internal margin of the dentaries. Abbreviations: dep, depression; idp, interdental plates; mg, Meckelian groove. Scale bar = 5 cm.

Figure 3 Surface model of the left dentary of SMA 0005 with anterior and posterior CT sections.

(A) Posterior end of the left dentary in lateral view, showing signs of alveoli of dentary teeth. (B) Anterior end of the left dentary in mirrored medial view, showing dense bone matrix with no sign of alveoli. Abbreviations: adt, artificial dentary tooth with an internal wire; alv, alveoli; db, dense bone. Scale bar = 5 cm.

Diagnosis. The absence of any traces of trauma, infection or healing indicates that the supposed pathology has happened long time before the death of the animal, possibly even during its early ontogeny. The compact bone in the anterior portion of left dentary further indicates that the first alveoli may be reduced during the healing response to merely externally visible, shallow pits, i.e., that the anterior part of the left dentary was edentulous. However, one has to keep in mind that the cause of the anterior medial depression is not clear and an alveolus nature cannot be ruled out entirely.

Cervicals

Fourth cervical

Description. The fourth cervical shows a conspicuous, irregularly shaped proliferation of bone originating from the posteromedial side of the left prezygapophysis (Figs. 4A and 4B, see also Fig. S2 in the Supplemental Information 1). The proliferation is anteroventrally and medially directed and measures c. 9 mm anteroposteriorly, c. 18 mm dorsoventrally and 25 mm lateromedially in its maximum extent. Anteriorly, the proliferation flattens and expands laterally, contacting the ventromedial side of the left prezygapophysis, so that it looks inverted L-shaped from dorsal view. From anterior view it is kidney-shaped with the concave edge facing ventrally. The surface of the structure is overall rugose. A further small anomaly is present medial to the left lateral margin of the spinopostzygapophyseal fossa above the neural canal (Figs. 4C–4E). The structure is posteroventrally directed and tapers distally. It measures c. 6 mm anteroposteriorly, c. 9 mm dorsoventrally and c. 11 mm lateromedially. Both structures show no signs of traumatic or infectious lesions.

Figure 4 Fourth cervical of SMA 0005.

(A) Fourth cervical in dorsal view, showing a pathologic exostosis (possible osteochondroma) between the prezygapophyses. (B) Possible osteochondroma from anterior view marked by a dotted line. (C) Fourth cervical in posterior view, showing another exostosis (possible inflammatory ossification) above the neural canel. (D) Possible inflammatory ossification (dotted line) in close-view. (E) Possible inflammatory ossification (dotted line) in posterolateral view. Abbreviations: ep, epipophysis; ex, exostosis; nc, neural canal; ns, neural spine; poz, postzygapophysis; prz, prezygapophysis; tp, transverse process. Scale bar = 5 cm.

Diagnosis. A clear diagnosis of both exostoses is difficult. As no external indicator of is observable, and as the structures belong to none of the regular parts and processes of the vertebra, the most plausible explanation could be an enthesopathy (= inflammatory ossification of ligamentous or muscular attachments), or an osteochondroma (= benign bone tumor). Here, the irregular shape of the large anterior exostosis may correspond with the cauliflower-like morphology of an osteochondroma (Murphey et al., 2000). However, the most conservative classification of this pathology would be idiopathic, as no cause can be ascertained.

Fifth cervical

Description. The neural arch of the fifth cervical shows a severe pathology at the base of the left postzygapophysis (Fig. 5, see also Figs. S3 and S4 in the Supplemental Information 1). In external view, a fracture runs around the whole process, indicating the complete rupture of the postzygapophysis. This fracture is surrounded by a large callus on the dorsolateral side, which gives the left postzygapophyseal pedicle a swollen appearance, and follows roughly the course of the epipoprezygapophyseal lamina. While the broken postzygapophyseal fragment seems secondarily well connected to the neural arch medially, the fracture line appears as a gap laterally, separating the callus in an anterior and posterior part, which are not connected to each other. The anterior part of the callus measures c. 38 mm. Anteriorly, it ends at the level of the posterior edge of the transverse process. The callus shows a stronger lateral (c. 14 mm) than dorsal expansion (c. 8 mm). The posterior part of the callus is smaller and measures c. 19 mm in its anteroposterior dimension. The external surface of both parts of the callus is smooth, without any rugosity. However, CT data reveal that the pathologic postzygapophysis consists mostly of dense, homogenous bone tissue, while the right postzygapophysis shows signs of large, internal cavities (Fig. 6) probably related with the pneumatization of the neural arch (see O’Connor, 2006).

Figure 5 Fifth cervical of SMA 0005.

(A) Fifth cervical in left lateral view, showing the callus and the fracture (dotted line) at the base of the left postzygapophysis. (B) Fifth cervical in lateral view in posterolateral view. (C) Callus (dashed line) and fracture (dotted line) in laterodorsal view. (D) Callus (dashed line) and fracture (dotted line) in lateroventral view. (E) Callus (dashed line) and fracture (dotted line) in oblique posterolateral view. Structures of minor interest in the neural arch are colored transparently for cover. Abbreviations: ca, callus; ep, epipophysis; ns, neural spine; poz, postzygapophysis; pp, parapophysis; prz, prezygapophysis; tp, transverse process. Scale bar = 5 cm.

Figure 6 CT section through the callus of the fifth cervical of SMA 0005.

(A) Fifth cervical in left lateral view, showing the progression of the CT section through the callus. (B1) Surface model of the fifth cervical in posterior view, showing the section through the neural arch at the anterior level of the callus. (B2) Tomogram of the anterior section. (C1) Surface model of the fifth cervical in posterior view, showing the section through the neural arch at the posterior level of the callus. (C2) Tomogram of the posterior section. The CT data show that the callus at the left postzygapophysis is made of dense bone, while the right postzygapophysis is pneumatised by a large internal cavity. Abbreviations: ca, callus; L, left side; ns, neural spine; poz, postzygapophysis; pn, pneumatization; prz, prezygapophysis. Scale bar = 5 cm.

Diagnosis. Based on the morphology, it seems that the pathology in the left postzygapophysis represents a traumatic fracture, which shows typical callus healing. This healing response seems to affect the pneumatization pattern of the postzygapophysis due to reactive bone growth.

Presacral ribs

Description. Several ribs of SMA 0005 show evidence for lesions. In the cervical region, one pathologic rib is found on the right side of the fourth cervical vertebrae (Fig. 7A, see also Fig. S5 in the Supplemental Information 1). In the dorsal region, the fifth rib from the right body side shows a fracture in its distal third. On the left body side, the third, seventh and ninth dorsal ribs show clear evidences of fractures, appearing all in the distal half of the rib, almost on the same level (Fig. 7B, see also Fig. S5 in the Supplemental Information 1). Other ribs from the left side also show deformations on this level. However, as clear fractures cannot be observed, a distinction between a pathologic or taphonomic origin is not possible for the deformed elements. All fractured bones identified show a distinct overlapping connection of both broken elements, sometimes with a slight displacement of the distal end. In some ribs minor callus formation is present.

Figure 7 Pathologic ribs and cortical traces in SMA 0005.

(A) Fourth cervical rib from the right side with fracture (dotted line). (B) Seventh dorsal rib from the left side with fracture (dotted line). (C) Cortical traces in the right ischium. (D) Cortical traces in the left scapula. Abbreviations: ca, callus. Scale bar = 2 cm.

Diagnosis. The observations of the fracture morphology in the ribs are consistent with the morphology of pseudarthroses (Cleas, Wolf & Augat, 2000; Loboa, Beaupré & Carter, 2001; Klein et al., 2003), which result from traumatic events. In some dorsal ribs the pseudarthroses are associated with small calluses, which are remains of the initial healing response of the fracture.

Scapula

Description. A complete, transverse fracture occurs in the proximal part of the left scapula (Fig. 8, see also Fig. S6 in the Supplemental Information 1). This fracture does not show a regular callus, but some osseous connection of the fractured end to the respective other fragment is apparent. Thus, the proximal fragment, which articulates with the coracoid and the humerus, is laterally displaced so that it sits on the distal fragment. The periphery around the distal end of the proximal fragment shows a rough striation in line with the longitudinal axis of the scapula. Furthermore, the proximal fragment is also displaced in that it is rotated ventrally, resulting in an artificial tilt between both fragments. The maximal overlap between the proximal and distal element is c. 81 mm. CT data of the specimen show that most of the overlapping parts only lie on top of each other, and that only the fracture ends constitute a fused bridge between the fracture elements (Fig. 8D).

Figure 8 Pathologic scapula of SMA 0005.

(A) Left scapula in anterolateral view, showing the fractured area of the scapula blade (dashed line). (B) Fracture (dotted line) in dorsal view. (C) Fracture (dotted line) in ventral view. (D) Tomogram of the scapula, showing that only the fracture ends constitute a fused bridge between the fracture elements (arrows), which is consistent with a pseudarthrosis. Abbreviations: co, coracoid; cf, coracoid foramen; fr, fracture; gl, glenoid facet; sc, scapula blade. Scale bar = 5 cm.

Diagnosis. The overall morphology and the CT data of the fracture are consistent with the morphology of a pseudarthrosis (Cleas, Wolf & Augat, 2000; Loboa, Beaupré & Carter, 2001; Klein et al., 2003), in which the lateral displacement of the proximal fragment may be explained by a number of reasons, including mechanical instability due to the morphology of the scapula as a blade-like element or the nature of the traumatic event that caused the fracture. One possible explanation might be found in the separation of muscle groups on the lateral scapula in a distal part (M. deltoideus scapularis) and a more proximal part (Mm. scapulohumeralis, M. deltoideus clavicularis), with the boundary between these two regions apparently coinciding with the area of the break in SMA 0005 (see Remes, 2008; Burch, 2014). Thus, the pull of the M. deltoideus scapularis would have rotated the distal end of the scapula outwards in respect to the proximal end, possibly accounting for the overlap. In contrast, the ventral rotation of the proximal fragment is probably caused by the mechanical load of the arm, pulling the fragment it is articulated with down. The malunion of the fracture fragment, once achieved, can hardly be reversed, and the weight of the attached arm together with movements induced by arm use and torso movements related to locomotion of the animal account for a lack of stabilization of the fracture.

In a preliminary report (Evers et al., 2013), we described a second potential fracture of the scapula in the distal part of the bone. However, a re-examination of the specimen suggests rather a post-mortem plastic deformation of this structure.

Humerus

Description. The left humerus of SMA 0005 shows an abnormal ulnar condyle (Fig. 9A, see also Fig. S7 in the Supplemental Information 1). It is elongate and thin, contrasting the more rounded morphology usually seen in theropods and also in the right element (Fig. 9B). Furthermore, no remains of prominent, tapered epicondyle can be found on the anterior side. The condyle has an irregular surface texture of numerous depressions of varying depth, as well as a deep oblique groove toward the anterior aspect of the medial side, which measure c. 50 mm in maximum length. Additionally, the ventral surface of the ulnar condyle bears some sharp, trough-like marks, measuring c. 20 mm in transversal length.

Figure 9 Left and right humerus of SMA 0005.

(A) Distal portion of the left humerus in anteromedial view, showing idiopathic pathologies. The pathologies contain an irregular cortical texture with numerous depressions (arrows), a deep oblique groove toward the anterior aspect of the medial side (dotted lines), and two sharp, trough-like marks on the ventral surface of the ulnar condyle (dashed lines). (B) Distal portion of the right humerus in anterior view. Abbreviations: epc, epicondyle; hu, humerus; ul, ulna; uc, ulnar condyle. Scale bar = 5 cm.

Diagnosis. The abnormal form and texture of the left ulnar condyle is interpreted as an idiopathic pathology, although the depressions might indicate a potential infection, an avulsion of ligaments or tendons (traumatic), a developmental disorder in ossification, or even a post-mortem modification.

Ischium

Description. The right ischium exhibits an oblique fracture located at a midshaft position (Fig. 10, see also Fig. S8 in the Supplemental Information 1). The distal fracture fragment sits on the medial side of the proximal fragment, and is slightly medially rotated. Due to this orientation, there is a distally widening interfragmentary gap between the distal end of the proximal fracture fragment and the proximal part of the distal fracture fragment, which can be best seen in anterior view (Fig. 10B). Consequentially, the end of the proximal fragment forms a laterally projecting tip. The gap is partially filled with matrix, which shows that is was internally not closed by connective tissue when the animal died. The fracture line can be traced almost around the entire shaft of the ischium (Figs. 10B–10E). However, toward the proximal end of the fracture, the fragments are well connected on the anteromedial side.

Figure 10 Ischium of SMA 0005.

(A) Both Ischia in anterolateral view, showing the oblique fracture at the shaft of the right ischium. (B) Interfragmentary gap (arrow) of the right ischium in anterior view. (C) Fracture (dotted line) of the right ischium in anterolateral view. (D) Fracture (dotted line) of the right ischium in lateral view. (E) Fracture (dotted line) of the right ischium in posterior view. Abbreviations: fr, fracture; ib, ischial boot; ip, ischial peduncle; op, obturator process; pp, pubic peduncle. Scale bar = 5 cm.

No sign of callus is visible around the pathologic structure. However, on the anterior side of the fractured area, the cortical surface is disturbed by a large trace of small, interconnected depressions with irregular size, shape, and depth. The traces continue well beyond the fracture almost to the ischial boot.

Diagnosis. Because no clear callus structure is visible, and the present fracture line is at no point bridged by cortical bone, it is possible that either a trauma caused an incomplete fracturing, but no healing took place (which would indicate a trauma-related death of the animal), or that breakage of the bone occurred post-mortem. Another possibility is that the fracture was complete, and that the anteromedially located connection of the fragments was secondarily achieved. In this case, the structure would fulfill the criteria of a developing pseudarthrosis (see Cleas, Wolf & Augat, 2000; Loboa, Beaupré & Carter, 2001; Klein et al., 2003). However, the nature of the bone around the anteromedially located connection described above is obscured by traces penetrating the surface (see discussion). These traces are found on the anterior side of the ischium and could be related to bone infection. However, as similar traces appear also on the surface of other bones, we would rather interpret this structure as most probably taphonomic in origin (see discussion, Figs. 7C and 7D).

Additional note. The left ischium shows a slight swelling and associated possible fracture line at the same level as the right element (Fig. 10A). However, parts of the possible pathology are obscured by a reconstruction of the lateral bone surface in this part. Thus, no detailed comments on the morphology of the potential abnormality and potential healing responses can be made. Due to the present uncertainty regarding this structure, we avoid further interpretation.

Foot

Left pedal phalanx II-2

Description. The pedal phalanx II-2 of the left foot has a bulbous callus covering about two-thirds of the element (Figs. 11A–11D, see also Fig. S9 in the Supplemental Information 1). The callus is located at the proximal part of the phalangeal shaft, and does not reach both the proximal and distal articulation facets. In the mid-shaft area, the callus surrounds the phalanx body almost entirely. Toward the proximal articulation, the callus forms a groove-like channel with a sharp and step-like edge, which circumferences the medial, dorsal, and lateral parts of the callus. Towards the distal articulation, the callus is laterally complanate and hence approaching the regular morphology again, while the medial aspect is strongly swollen. The ventral side of the proximal end is also strongly inflated, exhibiting a c. 5 mm thick bulge. The surface of the callus is generally irregular, while the degree of irregularity is reducing towards the distal part of the element, and weaker developed than in comparative specimens (Hanna, 2002; see discussion).

Figure 11 Pathologic phalanges in SMA 0005.

(A) Left pedal phalanx II-2 from lateral view, showing the callus and the reduced extensor tubercle. (B) Left pedal phalanx II-2 from ventral view, showing the callus and multiple small depressions (arrows). (C) Left pedal phalanx II-2 from medial view, showing the callus and two large depressions (arrows), possibly indicating a secondary infection of the bone. (D) Left pedal phalanx II-2 from dorsal view, showing the callus and one of the two large depressions on the medial side. (E) Right pedal phalanx II-2 from mediodorsal view, showing the normal condition and size of the extensor tubercle. (F) Left pedal phalanx IV-1 in lateral view, showing two idiopathic bulbous swellings (dotted lines). Abbreviations: ca, callus; et, extensor tubercle. Scale bar = 5 cm.

Compared to the non-pathological pedal phalanx II-2 of the right foot (Fig. 11E), the extensor tubercle is almost completely reduced in the left pedal phalanx II-2 with the most proximal point ending approximately 40 mm anteriorly in relation to the ventral flexor heel (see Figs. 11A–11D). However, the surface structure in the respective area of the phalanx is not indicative of taphonomic deformation or erosion. Therefore, this anomaly likely represents a pathologic structure, too.

Several depressions could be observed in this bone, penetrating the callus (Figs. 11B and 11C). The largest depressions appear posteromedially and are several millimetres deep. Here, the outer margin of the more posterior depression measures c. 6 mm by 10 mm and faces posteromedially. The second depression lies anteroventrally in respect to the former, and faces medially. Its outline is circular and measures c. 7 mm by 7 mm. Both depressions possess a distinct rim. The ventral aspect of the callus also shows several small round to oval-shaped depressions, which look very similar to the structures found in the left humerus (see Fig. 9A).

Diagnosis. The presence of a callus indicates response to a traumatic pathology, which may have been caused in various ways, including a traumatic accident or constant overloading of the element during locomotion (see discussion). The absence of the extensor tubercle is conspicuous, and possibly but not necessarily related to the pathology that caused the callus. It is possible that the extensor tubercle was never fully developed, which would point to a developmental pathology. Alternatively, the extensor tubercle could be lost during remodeling of bone in relation to advanced callus healing (see discussion). Finally, the depressions found in the medial and ventral surface of the phalanx may underlie a pathologic origin. If this is the case, their morphology is most consistent with lesions caused by osteomyelitis (Ortner & Putschar, 1981; Pschyrembel, 1990; Rothschild & Martin, 2006). However, because the simple morphology of the small depressions on the ventral side not unequivocally linkable to an infectious cause, a possible taphonomic origin should not be ruled out.

Left pedal phalanx IV-1

Description. In the left pedal phalanx IV-1, there are two bulbous swellings on the lateral side (Fig. 11F see also Fig. S9 in the Supplemental Information 1). One is positioned underneath and posteroventral to the lateral ligament pit, and another one is situated at the posterolateral side near the proximal articulation. The anterior swelling follows the slightly sinuous curvature of the ventral side of the bone in lateral view, which is the result of the constricted phalangeal shaft between the proximal and distal joints, which are both dorsoventrally expanded in relation to the shaft. The posterior swelling parallels the posterolateral and lateral margin of the proximal articulation, and is therefore vertically oriented. The swellings are separated by a small oblique gap, under which the bone seems to have retained its usual form. Both swellings have a smooth surface structure not different from other parts of the bone, but are not found on the same element of the right foot and are therefore abnormal. The swellings are different from the callus on the left pedal phalanx II-2, as they have a clearly delimited and abrupt border to either side.

Diagnosis As no lesions or external fracture lines are present, we classify the pathology on the left pedal phalanx IV-1 as idiopathic.

Discussion

Identification and cause of pathologies

According to the scheme of Hanna (2002), the pathologic elements of SMA 0005 can be classified as follows: the fifth cervical vertebra, the scapula, several ribs and right ischium are probably traumatic, and the callus structure of the left pedal phalanx II-2 is probably traumatic-infectious. In contrast, the supposed pathologies in the lower jaw and in the reduced extensor tubercle of the left pedal phalanx II-2 cannot be assigned to a certain type of this scheme, as they show evidence of advanced healing. They are most likely traumatic or developmental in origin. The same is true for the abnormal outgrowths in the neural arch of the fourth cervical, which are most likely developmental in origin or idiopathic. The pathology on the left humerus is interpreted as traumatic, developmental, infectious or idiopathic, whereas the left pedal phalanx IV-1 is classified as idiopathic. With exception of the ischium, all lesions interpreted as traumatic/traumatic-infectious pathologic elements show unambiguous evidences of healing, indicating that the respective pathologies did not cause the death of SMA 0005. The role of the ischial fracture as a possible cause of death will be discussed below.

The deformed anterior end of the left dentary of SMA 0005 is most likely pathologic, but no obvious lesions are developed. This indicates that the supposed pathology was probably completely healed and happened long before the animal died. A pathology in the anterior part of the dentary is also found in the Allosaurus specimen USNM 2315 (Gilmore, 1920; Tanke & Currie, 2000; Molnar, 2001). The symphysial region is strongly deformed, leading to a concavity in the anterior part of the dentary, which is bordered anteriorly by a dorsally pointing, hook-like projection. The anterior alveoli seem completely resorbed so that the symphysial region is edentulous. According to Gilmore (1920) and Tanke & Currie (2000), the anterior end of the dentary in USNM 2315 was probably bitten off and then heavily remodelled during the healing process. As no sign of other pathologic deformations are visible in the anterior end of the dentary, the supposed trauma happened probably long before the death of the animal. Based on the CT data, a similar condition might be also present in SMA 0005, as no traces of alveoli could be detected in the anterior most region of the left dentary. At the current stage it is not clear, if the medial depression found at the anterior end of the dentary might represent a strongly modified alveolus. Assuming this possibility, however, it is still questionable if the structure could actually produce teeth, because its morphology indicates a potential damage of the tooth anlagen on the medial side. Assuming a similar scenario for the origin of the pathology in SMA 0005 and USNM 2315, both specimens may indicate face-biting behavior in Allosaurus, which was previously also hypothesized for other large-bodied theropods (e.g., Sinraptor, Albertosaurus, Daspletosaurus, Gorgosaurus and Tyrannosaurus), including juvenile specimens in some of these taxa (Tanke & Currie, 2000; Peterson et al., 2009; Bell, 2010; Hone & Tanke, 2015). Another example of a remodeled alveolus with possible traumatic origin was recently described for a single maxilla of the basal tetanuran Sinosaurus (= “Dilophosaurus sinensis”) (Xing et al., 2013). However, it is also possible that the abnormal shape of the dentary in SMA 0005 results from developmental malformation or a fracture that happened in earlier ontogeny, which left no remaining traces.

The deformation of the anterior end of the dentary has implications for the structure and function of the mandibular symphysis in Allosaurus. As pointed out by Holliday & Nesbitt (2013), most basal theropod dinosaurs have a very simple mandibular symphysis that consists of a simple flattened medial area of the anterior end of the dentary. However, even in such an osteologically simple structure the actual union of the left and right mandible by connective tissue can be quite variable (see Holliday et al., 2010). The deformation of the left, but not the right mandible in SMA 0005 indicates that there was no very tight junction between the two mandibular rami, and the symphysis and the jaws as a whole functioned despite the different morphologies and resulting differences in the level of the alveolar margins in the left and right mandible. This is supported by the deformation seen in USNM 2315, which also affected the mandibular symphysis.

Another interesting aspect of the pathologic dentary of SMA 0005 is its similarity with dentaries of spinosaurid megalosaurs. As there are no direct indications of pathology in the bone itself, and the pathologic nature can only be inferred by comparison with the other dentary, which shows a more typical morphology for Allosaurus. This element, if found isolated, would probably not have been classified as Allosaurus. This has previously happened with the dentary USNM 2315, which was originally described as a new species, Labrosaurus ferox, by Marsh (1884). Thus, caution is needed when evaluating the systematic position of isolated elements to rule out possible pathologies.

The most common pathology in the axial skeleton of dinosaurs is the fusion of single vertebrae, which often appears in the caudal series. Possible causes of vertebral fusion are e.g., congenital abnormality (e.g., Witzmann et al., 2008), infections (Rothschild, 1997; Rothschild & Martin, 2006), malformations during the healing process of a trauma (Rothschild, 1997; Butler et al., 2013), diffuse idiopathic skeletal hyperostosis (DISH) (Rothschild, 1987; Rothschild & Berman, 1991) or spondyloarthropathy (Rothschild & Martin, 2006; Witzmann et al., 2014). No evidence of vertebral fusion is found in SMA 0005.

Like in the Allosaurus specimen MOR 693, the dorsal neural spines of SMA 0005 show irregular-shaped exostoses, which were diagnosed by Hanna (2002) as idiopathic pathological ossification of interspinous ligaments. However, little research has been done on the classification of ossified ligaments and other soft tissues in dinosaurs. In ornithopods and dromeosaurid theropod dinosaurs, ossified tendons are found to stiffen parts of the axial skeleton, and these structures are commonly not interpreted as pathologic (e.g., Ostrom, 1969; Norell & Makovicky, 1999; Organ, 2006). However, in many theropod dinosaurs, rugose outgrowths are found on the anterior and posterior sides of the neural spine, which are thought to be part of ossified prespinal and postspinal ligaments, respectively. These structures occur more frequently on larger specimens (e.g., Allosaurus BYU 725/12901, BYU 725/12902, BYU 725/13051, UMNH VP 8365, UMNH VP 13813; Acrocanthosaurus SMU 74646, Harris, 1998; cf. Spinosaurus BSPG 2006 I 57; Majungasaurus UA 8678, O’Connor, 2007), although there are also smaller specimens with such ossification (e.g., Allosaurus UMNH VP 7341, DINO 11541; Dahalokely UA 9855, Farke & Sertich, 2013). However, in individuals preserving an articulated or associated vertebral series, no clear pattern of intervertebral tendon ossifications can be observed in contrast to e.g., ornithopods (Organ, 2006). In Neovenator, the posterior cervical vertebrae show ossifications at the apexes of the neural spines, and most dorsal vertebrae show such structures (Brusatte, Benson & Hutt, 2008), while in Baryonyx, a mid-cervical vertebra (BMNH R9951) shows relatively large ossifications, whereas more posterior positioned vertebrae lack such structures and only show rugose attachment sites for the respective ligaments on the neural arch (Charig & Milner, 1997). In some cases, the interspinal ossifications have been suggested to be of diagnostic and thus taxonomic value (Chure, 2000). The above cases show that ligament ossifications in dinosaurs are frequently not interpreted as pathologic, and because many theropods show ossifications of at least the attachment areas of interspinal ligaments, we advocate that they should be regarded as non-pathologic, pending more detailed research on the topic.

However, the exostoses found in the fourth cervical of SMA 0005 differ in their position and morphology from the examples mentioned above. Here, the strongly irregular shape of the anterior exostosis resembles the morphology of an osteochondroma, which represents the most common type of bone tumors in humans (Murphey et al., 2000; Sekharappa et al., 2014). In captive wild extant mammals and reptiles (including birds), however, the development of tumors is rather rare (Ratcliffe, 1933; Effron, Griner & Benirschke, 1977; Huchzermeyer, 2003). Thus, it is not surprising that the unambiguous diagnosis of tumors in dinosaurs is limited to only a few cases (e.g., Rothschild et al., 1998; Rothschild, Witzke & Hershkovitz, 1999; Rothschild et al., 2003; Arbour & Currie, 2011; Rega, 2012). Due to the restricted knowledge of tumor formation in dinosaurs in general, this diagnosis has to be seen with caution. However, if the diagnosis is correct, the anterior exostosis found in the fourth cervical of SMA 0005 represents the third case of an osteochondroma in dinosaurs (Rega, 2012). The smaller, more regular-shaped exostosis on the posterior side of the neural arch does not fulfil the criteria for a bone tumor. One possible explanation for these structures could be an inflammatory ossification of the ligamentum elasticum interlaminare, which attaches right above the neural canal of cervical vertebrae (Tsuihiji, 2004), probably affecting the neck mobility. However, if none of the presented diagnosis is correct, both exostoses have to be classified as idiopathic.

Evidence for traumatic pathologies in the vertebral column is also rare in dinosaurs. Carpenter et al. (2005) describes an anterior caudal of Allosaurus with a possible puncture in the left transversal process, which was most likely injured by a Stegosaurus tail spike, indicating a predator–prey relationship between both dinosaurs. Traumatic caudals found in the basal sauropodomorph Massospondylus (Butler et al., 2013) and the hadrosaur Edmontosaurus (Carpenter, 2000) probably result from unsuccessful attacks of large-bodied theropods, indicating active hunting behavior in the latter. Tail injuries were also described in the theropod dinosaur Majungasaurus as well as in many ceratopsian and hadrosaurian dinosaurs, but their origins remain speculative (Farke & O’Connor, 2007) or are interpreted as result of intraspecific interactions, such as accidently tail trampling due to herding behavior (Tanke & Rothschild, 2010; Tanke & Rothschild, 2014). In contrast, the supposed traumatic fracture found in the fifth cervical of SMA 0005 probably results from a serious accident. Although the whole left postzygapophysis was basically broken, the lesion shows evidence of healing in form of a callus, indicating the survival of the accident. A possible explanation for such a rather unusual break might be found in the importance of the neck in hunting behavior in large theropods (e.g., Snively & Russel, 2007a; Snively & Russell, 2007b; Snively et al., 2013). Thus, the injury might have resulted from a failed hunting attack or from struggling prey, in which case this represents further evidence for active hunting in Allosaurus (see also Carpenter et al., 2005). However that may be, the severity of the injury most probably had a serious effect on the neck mobility of the specimen (see Snively et al., 2013).

Fractured or infected presacral ribs are one of the most common pathologies found within theropods (Molnar, 2001), in which, however, cervical ribs are less affected than dorsal elements. Pathologic cervical ribs are reported for Megalosaurus (Tanke & Rothschild, 2002), Allosaurus (Petersen, Isakson & Madsen, 1972) and Tyrannosaurus (Brochu, 2003), whereas corresponding dorsal rib pathologies are found in various large-bodied theropods like the abelisaurid Majungasaurus (Farke & O’Connor, 2007), the allosauroids Acrocanthosaurus (Harris, 1998), Allosaurus (Molnar, 2001; Hanna, 2002; Rothschild & Tanke, 2005; USNM 4734, T Holtz, pers. comm., 2014), Mapusaurus (Bell & Coria, 2013) and Sinraptor (Currie & Zhao, 1993) and the tyrannosaurids Albertosaurus (Bell, 2010), Gorgosaurus (Lambe, 1917) and Tyrannosaurus (Brochu, 2003; Rothschild & Molnar, 2008). The examples mentioned above show different kinds of pathologies, i.e., trauma-related callus formations (Harris, 1998; Hanna, 2002; Brochu, 2003), pseudarthroses (Harris, 1998; Brochu, 2003; Rothschild & Molnar, 2008; Bell, 2010) or lesions by microbial infections (Harris, 1998; Hanna, 2002; Brochu, 2003; Bell & Coria, 2013), in which the latter could be the result of secondary infections of the injury. Hanna (2002) further describes the formation of idiopathic spiculae on two fractured dorsal ribs in the Allosaurus specimen MOR 693. In SMA 0005, all pathologic ribs show evidence of lesions, which are probably traumatic-related pseudarthroses. Here, the pseudarthrosis as a healing response (rather than callus healing) in the cervical rib results most likely from regular neck movements (Snively & Russel, 2007a; Snively et al., 2013), whereas the pseudarthroses found in the dorsal ribs were probably caused by constant movement of the ribcage during breathing (Claessens, 2009a; Claessens, 2009b) or due to thorax movements during locomotion (see Mallison, 2010).

The fractured scapula shows a clear case of a pseudarthrosis as healing response, which resulted from the apparent malunion of the fractured elements. Mechanical loading is additionally likely, as the proximal fragment, which is articulated with the rest of the arm, is tilted ventrally. The extent of the malunion may be indicative of syn-traumatic displacement, which potentially indicates great destructive force acting upon the element. Accordingly, the left arm in SMA 0005 was likely dysfunctional after the trauma. Other examples of pathologic scapulae in theropods can be found in the Allosaurus specimen USNM 4734 (Gilmore, 1920; Rothschild, 1997; Molnar, 2001) and in Yangchuanosaurus (Xing et al., 2009). The scapula of USNM 4734 shows a strong, arched dislocation between both fragments, in which the proximal element developed a spine-like exostosis on the ventral margin of the projecting portion of the proximal fragment (Gilmore, 1920; Rothschild, 1997). In contrast, the injury of the scapula in Yangchuanosaurus shows callus formation as healing response (Xing et al., 2009), indicating an incipient fracture. Other examples of pathologic scapulae seem not to be related to traumatic events, but with the development of exostoses (e.g., Acrocanthosaurus (NCSM 14345, C Foth, S Evers & O Rauhut, pers. obs., 2012) and Neovantor (Brusatte, Benson & Hutt, 2008)), idiopathic lesions (e.g., Allosaurus (MOR 693; Hanna, 2002)) and infectious lesions in relation to osteomyelitis (e.g., Allosaurus (UUVP 1528, UUVP 5599, Molnar, 2001; Hanna, 2002)).

Because the injuries of the left scapula and the dorsal ribs from the left side are present at almost the same level of the thorax, it is possible that these traumas happened in one single event, e.g., a serious fall, or a defensive blow from a sauropod tail. This scenario would be even more probable if the deformations found in the other dorsal ribs from the left side have a traumatic origin, too. However, as stated above, this cannot currently be confirmed, as they cannot be distinguished from taphonomic deformations. Multiple rib fractures from one thorax side are also documented in Acrocanthosaurus (Harris, 1998), Allosaurus (Hanna, 2002; USNM 4734, T Holtz, pers. comm., 2014) and Tyrannosaurus (Brochu, 2003), possibly also resulting from one single traumatic event.

The most complex pathology appears in the left pedal phalanx II-2, including a reduced extensor tubercle, a callus formation of the phalangeal shaft, and several depressions penetrating the callus. The small size of the callus and the absence of any external fracture lines in the left pedal phalanx II-2 maybe indicate that the bone was not injured in an actual accident, but affected by chronic traumatic stress, resulting in a stress fracture as it is described for various dinosaurs. The origin of the reduced extensor tubercle remains speculative, possibly being developmental in origin or resulting from a trauma that healed long before the animal’s death, in which the tubercle was lost during resorption of bone during advanced healing stages. As the extensor tubercle in its normal condition should prevent the hyperextension of pedal phalanges, the absence of this process may have led to a frequent overloading of pedal muscles and ligaments in SMA 0005. Thus, it is possible that this pathology is physically linked to the callus formation in the proximal portion of the phalangeal shaft, as frequent hyperextension may have caused chronic traumatic stress to the phalanx. In a different scenario, the callus is a response to a fracture or stress fracture unrelated to a developmental disorder of the extensor tubercle. At any rate, stress fractures in pedal phalanges are a type of pathology that commonly occurs in theropods (Madsen, 1976; Rothschild, 1988; Rothschild, Tanke & Ford, 2001; Rothschild & Tanke, 2005; Farke & O’Connor, 2007; Bell, 2010; Zanno et al., 2011; Anné et al., 2014), and are thought to be related to strenuous activities (Rothschild, Tanke & Ford, 2001; Rothschild & Tanke, 2005).

The additional penetration of the callus by several depressions indicate a potential secondary infection of the pedal phalanx, perhaps caused by a syn-traumatic injury of adjacent soft tissue, through which microbial pathogens got access to the bone and cause contiguous osteomyelitis. Thus, the callus pathology is most likely traumatic-infectious. Secondary infections of callus structures as well as infections not clearly linkable to fractures seem to be common in pedal phalanges of Allosaurus (MOR 693, UUVP 1657, UMNH VP 6295, UMNH VP 6284, UMNH VP 10755, UMNH VP 6287, UMNH VP 6299). In two specimens (MOR 693; UUVP 1657), the supposed secondary infections led to colossal exostoses, causing chronic pain and restriction in the locomotion. However, the medial and ventral depressions of the left pedal phalanx II-2 of SMA 0005 differ distinctly in their morphology. Thus, their origin does not have to be necessarily linked to each other. Especially, due to the inconspicuous morphology of the small depressions on the ventral side of the phalanx a taphonomic origin cannot be ruled out with complete certainty.

The cause for the abnormalities of the left humerus and left pedal phalanx IV-1 are unknown, and could be potentially infectious, traumatic or developmental. The small depressions on the anterior side of the ulnar condyle of the humerus could be potentially related to post-mortem modification.

The most severe and potentially fatal pathology occurs in the ischium, which most likely represents a traumatic fracture. Pathologies in the pelvic region are not often documented in theropods and usually restricted to the ilium (Molnar, 2001; Hanna, 2002; Bell & Coria, 2013). In the Allosaurus specimen UUVP 5985, the ilium is fused with the ischium (Hanna, 2002). However, an ischial fracture is to our knowledge not documented within theropods so far. The fracture of the right ischium exhibits a large interfragmentary gap with a projecting fragment on the lateral side. Because unambiguous healing responses are absent around the fracture, the possibility that the ischium was broken post-mortem has to be considered. However, scenarios in which a skeletal element with a designated long axis fractures in the oblique way described above are hard to come by, and we think the most parsimonious explanation for the observed fracture is a traumatic event during life. This is perhaps supported by the presence of sandy sediment matrix in the interfragmentary gap, as a void would be expected to be filled by different material if the fracture was the result of stress related to tectonic activity. Although no callus structure is found around the fracture, its absence per se cannot be seen as a clear indicator for the lack of a healing response, as the integration of the ischium in a complex network of locomotor musculature (Carrano & Hutchinson, 2002; Hutchinson et al., 2005) would predict intense motion along the fracture, favoring the formation of a pseudarthrosis (Cleas, Wolf & Augat, 2000; Loboa, Beaupré & Carter, 2001). As seen in the scapula, large parts of the fracture line can remain unfused in a pseudarthrosis, and the fragments can be adhered at the end points of overlapping fracture fragments. In SMA 0005, the periphery of the connective bone in the scapula is structurally marked by fine striations and modifications from the smooth surface of healthy bone. Unfortunately, the irregular texture of the ischium, which we interpret as taphonomic (see below), prevents an assessment of the bone structure around the area where the ischium fragments meet, as the pattern would have overprinted the original bone surface structure. Therefore, it cannot be clarified if the connection of the fragments is the result of incomplete fracturing, or a secondary bridging due to a healing response. As experimental studies on animal fractures have shown that overhanging fracture ends tend to be resorbed during the healing process (Loboa, Beaupré & Carter, 2001), the presence of a laterally projecting fragment indicates that the healing process, if already started, was still in an early phase, supporting the hypothesis that the supposed trauma happened shortly before the animal died, and is accordingly a possible cause of death. It is likely that the locomotion ability of SMA 0005 was significantly limited or even inhibited by the injury, consequentially affecting life traits like its hunting success. The reason for the traumatic event remains speculative, although it must have been a forceful incident.

The irregular cortical texture found around the ischial fracture is probably not pathologic. The right pubis shows also large traces of similar structure. Smaller traces can be found in the right coracoid, both scapulae, the left humerus, the left ischium, the left pubis and the left fibula (Figs. 5C and 5D). The structures differ in their morphology from the supposed lesions found on the medial side of the in right pedal phalanx II-2, as they possess a very irregular outline with a weak margin and a complex inner topography, which is composed of interconnected round pits with irregular size and depth (c. 1 to 3 mm). This morphology is similar to the superficial pits found on various sauropod bones from the Morrison Formation, which are most likely taphonomic in origin (Fiorillo, 1998; Hasiotis, Fiorillo & Hanna, 1999). Possible causes for these traces are bone corrosion due to soil acidity (Fiorillo, 1998) or scavenging by insect larvae (Hasiotis, Fiorillo & Hanna, 1999).

Implications for paleobiology and lifestyle

The number of pathologic specimens in general and the number of pathologies within fairly complete Allosaurus individuals suggest that members of this taxon had an active lifestyle predisposed to injury. Most pathologies found seem to be traumatic in origin, but only few show evidence of secondary infection (Molnar, 2001). This either suggests that inflamed wounds quickly caused death, leaving no osteological traces, or that the immune defense of these animals was successful in prohibiting infections and the spread of such. Oftentimes injuries were indeed survived, as evidence for healing responses are abundant in the theropod fossil record. In previous studies (e.g., Hanna, 2002; Butler et al., 2013; Vittore & Henderson, 2013) mammalian immune response has often been used as a model for explaining pathologic structures thought to be related to tissue-invasive microbial infections in non-avian dinosaur taxa. This is because bone dynamics between both groups have been shown to be similar (Rega, 2012). However, similarities in bone dynamics do not necessitate similarities in immune response. While the mammalian immune response to infections usually is the formation of suppurative abscesses, extant reptiles (including birds) form small cysts of fibrin (fibriscesses) at the sources of infection, which tend to calcify in advance stages (Montali, 1988; Gomis et al., 1997; Huchzermeyer & Cooper, 2000; Cooper, 2005; Rega, 2012). In spite of evidence for severe infections in Allosaurus (e.g., pedal phalanges in Hanna, 2002), infections seem to be localized on single bones, and generally relatively rare in spite of the number and severity of pathologies found in this and other studies. This suggests that the spread of infection and therefore haematogenous osteomyelitis occurred rarely in Allosaurus, which is consistent with the reptilian immune response and the localization of pathogens by means of fibrin clotting. Therefore, applying the extant phylogenetic bracket, a reptile-like immune response should be suspected for tissue-invasive microbial infections in non-avian dinosaurs, too. Consequently, application of a mammalian model for infectious pathologies in non-avian dinosaurs should be avoided (see Arbour & Currie, 2011; Rega, 2012). As the localization of pathogens in fibriscesses successfully prevents haematogenous osteomyelitis in reptiles, the risk of lethal infections due to the spread to other body regions is minimized (Rega, 2012). This is supported by the fact that theropods show only very localized indications for infections (Molnar, 2001).

The severity of pathologies in SMA 0005 and other Allosaurus specimens (Gilmore, 1920; Molnar, 2001; Hanna, 2002) points to a frequent exposure to hazardous situations. This might be seen as evidence for an active predatory life style. If this is accepted, many of the traumatic pathologies found could be the result of hunting accidents (see e.g., Carpenter et al., 2005). Some of the pathologies seen in Allosaurus, like the broken cervical postzygapophysis and scapula of SMA 0005, the hypertrophied pedal phalanx of MOR 693 (Hanna, 2002) and UMNH 1657 (Madsen, 1976; Hanna, 2002), or the fibula of USNM 4734 (Gilmore, 1920) can be expected to severely limit the movement, manoeuvrability, and speed of the animals. This in turn should affect the hunting success, but also intra- and interspecific competition for various other resources (water, territories, captured prey and carrion) of such an individual, which would be expected to mean certain death within a relatively short period of time. Indeed, the broken ischium qualifies as a strongly limiting and severe injury, and is potentially related to the death of SMA 0005. However, the number of cases of advanced healing for severe injuries within various Allosaurus specimens (including SMA 0005) might corroborate the presence of an intermediate metabolic rates compared to that of ectothermic non-avian reptiles and endothermic birds and therian mammals (Grady et al., 2014; Werner & Griebeler, 2014), so that a potential lower nutrient demand (compared to endothermic animals) allowed the survival of the animal despite its injuries because of the reduced necessity to feed frequently. Moreover, the common survival of the injured individual could be further related to scavenging or gregarious behavior, in which nutrition supply does not rely on the hunting success of a single individual. Although stratigraphic and taphonomic information is not provided in detail for all Allosaurus remains, this taxon represents the most abundant theropod within the Morrison Formation, and is frequently found with several specimens within near proximity to one another (see Gilmore, 1920; Madsen, 1976; Foster, 2003; Loewen, 2009), supporting a possible gregarious behavior. Within theropod dinosaurs, similar behavior has been further hypothesized for the coelophysoids Coelophysis (Colbert, 1989) and Syntarsus (Raath, 1990), the carcharodontosaurid Mapusaurus (Coria & Currie, 2006), the ornithomimosaur Sinornithomimus (Kobayashi & Lü, 2003; Varricchio et al., 2008), and the tyrannosaurids Albertosaurus (Currie, 2000; Currie & Eberth, 2010), Daspletosaurus (Currie et al., 2005) and Tyrannosaurus (Larson, 2008), which were often found in (nearly) monodominant assemblages with various ontogenetic stages. Although monodominant assemblages are scarce in the Morrison Formation (Foster, 2003; Gates, 2005), at least the Cleveland-Lloyd Dinosaur Quarry is by far dominated by Allosaurus. In spite of repeated taphonomic and sedimentological investigations of the quarry (Bilbey, 1998; Bilbey, 1999; Gates, 2005), the abundance of Allosaurus has not been satisfactorily explained, so that a gregarious scenario should not be ruled out a priori at this point. However, frequent findings of several associated specimens of Allosaurus and other theropods in single localities could alternatively reflect aggregations around a large food resource (predator trap hypothesis; see Dodson et al., 1980; Bilbey, 1999) or a drought-induced mass accumulation (Gates, 2005; see also Wings et al., 2012). Nevertheless, future discoveries related to the metabolic performance and social behavior of Allosaurus and other non-avian theropod dinosaurs could potentially falsify or support these hypotheses, leading to a reinterpretation of the paleobiology and life history of this and other pathological individuals.

Conclusions

The Allosaurus SMA 0005 represents a further specimen of this taxon with multiple pathologies, which were mostly interpreted as traumatic in origin, pertaining to all body regions (i.e., skull, axial skeleton, pectoral and pelvic girdle, and extremities). Traces of healing responses in all pathologic bones but the ischium suggest the survival of possible accidents and infections, but also an active predatory lifestyle predisposed to injury. The scarcity and local restriction of infectious pathologies is in agreement with a reptile-like immune response preventing the spread of infections via the blood stream. The survival of injuries affecting the physical fitness in Allosaurus may indicate gregarious behavior. However, verification of this hypothesis would require more direct evidence, like an unambiguous find of a group or direct trackway evidence (e.g., McCrea et al., 2014). Alternative explanations for the frequent survival of traumas could be furthermore the presence of a metabolic rate below those of endothermic organisms or scavenging behavior. The probable fracture in the ischium was potentially fatal, as no advanced traces of healing could be identified.

Supplemental Information

Supplemental Information 1 Additional figures

This file contains figures from the main text without interpretive lines of pathologies for comparison.

Click here for additional data file.

We would like to thank Hans ‘Kirby’ Siber, Thomas Bolliger (both Sauriermuseum Aathal) and entire Aathal team for access to SMA 0005 and logistic support. We further thank Philipp Schütz (EMPA) for assisting during the CT scans, Randy Irmis and Carrie Levitt-Bussian (both Utah Museum of Natural History), Brooks Britt and Rodney Scheetz (both Brigham Young University), and Jack Horner and John Scanella (Museum of the Rockys) for access other Allosaurus material as well as Vince Schneider and Lindsay Zanno (both North Carolina Museum of Natural Sciences) for sharing pictures of Acrocanthosaurus, and Roger Benson (University of Oxford) for sharing pictures of Neovenator. Mark Loewen (Utah Museum of Natural History), Richard Butler (University of Birmingham), Judith Engmann (Medizinische Hochschule Hannover) and Walter Joyce (University of Fribourg) are acknowledged for fruitful discussions. Finally, we want to thank Andrew Farke (Raymond M. Alf Museum of Paleontology) and Tom Holtz (University of Maryland) for revising and Matthew Wedel (Western University of Health Sciences) for editing the manuscript, which helped to improve the final version.

Institutional Abbreviations

BSPG Bayerische Staatssammlung für Paläontologie und Geologie, München, Germany

BYU Earth Science Museum, Brigham Young University, Provo, USA

DINO Dinosaur National Monument, Vernal, USA

MOR Museum of the Rockies, Bozeman, USA

NCSM North Carolina Museum of Natural Sciences, Raleigh, USA

SMA Sauriermuseum Aathal, Switzerland

UA Université d’Antananarivo, Antananarivo, Madagascar

UMNH Natural History Museum of Utah, Salt Lake City, USA, (formerly UUVP, University of Utah Vertebrate Paleontology)

USNM National Museum of Natural History, (formerly United States National Museum), Smithsonian Institution, Washington, D.C., USA

Additional Information and Declarations

Competing Interests

Author Contributions

The authors declare there are no competing interests.

Christian Foth and Serjoscha W. Evers conceived and designed the experiments, performed the experiments, analyzed the data, wrote the paper, prepared figures and/or tables.

Ben Pabst conceived and designed the experiments, performed the experiments, analyzed the data, reviewed drafts of the paper.

Octávio Mateus conceived and designed the experiments, reviewed drafts of the paper.

Alexander Flisch and Mike Patthey conceived and designed the experiments, performed the experiments, contributed reagents/materials/analysis tools, reviewed drafts of the paper.

Oliver W.M. Rauhut conceived and designed the experiments, analyzed the data, wrote the paper, reviewed drafts of the paper.

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
