# Peer review of "New insights into the lifestyle of Allosaurus (Dinosauria: Theropoda) based on another specimen with multiple pathologies"

_PeerJ, doi:10.7717/peerj.940_

## Round 0.1 · original submission · Minor Revisions

This is a nice paper and it will be a valuable addition to the literature on dinosaur paleopathology and on the lives of dinosaurs generally. The reviewers' suggestions for improvement are all pertinent and reasonable. In revising the manuscript, please pay special attention to these areas:

1. Take care to separate the description of the pathological elements--which should be a comparatively dry statement of currently observable facts--from the interpretation of what caused the pathology (trauma, infection, etc.), which is a hypothesis about causal mechanisms. The section "Describing Bone Disease" in Rega (2012) provides some descriptive guidelines.

2. I strongly agree with Dr. Farke's suggestion to include unmodified photos alongside the traced-over or otherwise enhanced versions in your figures. Remember the old aphorism, "If your data speak for themselves, try not to interrupt." That's not to say that the marked-up photos aren't valuable--most readers won't have the opportunity to see the material first-hand and many of the features you describe may be difficult to discern without some pointers--but it's important to give readers the unmodified version so they can distinguish between the fossil and the interpretation. I know revising figures is hard work but this is a chance to add lasting value to the paper.

·

Basic reporting

- The figures and text may be improved in some areas; see general comments.

Experimental design

No concerns in this area.

Validity of the findings

I would recommend a more thorough separation of data and interpretation; see "General Comments"

Comments for the author

This paper by Foth and colleagues provides a nice overview of paleopathology in a single specimen of the carnivorous dinosaur Allosaurus. These kinds of data are useful to many paleontologists, particularly for inference of behavior as well as study of the evolution of disease processes. I would recommend a few particular areas of revision, including: 1) More intentional separation of data (what the specimen looks like) and interpretation (what processes caused the morphology); 2) Inclusion of information on what "normal" morphology looks like; and 3) Additional illustration of the morphologies described (perhaps using the CT scan data).

SPECIFIC COMMENTS

- I would be more cautious in use of various terms such as "trauma," "infection", and the like. These are a level of interpretation, and although probably good explanations for the observations presented here, do imply a level of certainty that may or may not be warranted. It is important to separate observation and interpretation throughout the paper. I might recommend presenting each anomaly as "lesion interpreted as..."

- In the introduction, Farke et al. 2009 might be a more appropriate citation than Farke 2004, because it more directly deals with paleopathology. Farke 2004 focuses more on modeling behavior with some minor reference to pathology; Farke et al. 2009 directly documents and interprets lesions on ceratopsian skulls. In the opening sentence, I should also note that Farke & O'Connor did not explicitly interpret the specimens they described as inter- or intra-specific interactions, but instead were fairly cautious about even going so far as to say the observed anomalies were traumatic in origin.

- Are the CT scan data archived anywhere? Even if not openly posted on a digital repository, I would advise that there be a mechanism for future workers to be able to track down the data. E.g., reposited at the SMA.

- p. 7: UMNH is the correct abbreviation, but the correct institutional name is now Natural History Museum of Utah. Confusing, but I'm told this is the way it is now.

- I would strongly recommend separating the "results" into a descriptive section and interpretive section. Thus, present all of your data in one paragraph (i.e., what do the anomalies look like), and then the interpretation of the data in the next (i.e., what is the diagnosis). Thus, even if interpretations change down the line (and they undoubtedly will), future readers can still easily get to the bare facts of the description.

- Unfortunately, the figures are rather hard to interpret, and in some cases it is very difficult to make out the exact lesions that are being discussed. In some cases, the abnormal areas are obscured by labeling, dashed or dotted lines delimiting the area of interest, or mottling in the bone surface color. To address this issue, I have several suggestions. First, consider including both an unmodified version of the photo side-by-side with a modified version (showing lines around the fracture callus, or whatever). A second possibility would be to include an unmodified photo next to a simple line drawing highlighting the pathological areas. A third possibility would be to include a photogrammetric or CT scan-based representation of the bones, to remove color and allow more control of the presentation. In some figures (e.g., Figure 4), it is quite difficult to distinguish areas of lesion from taphonomic cracks, etc.

- For the lower jaw, CT scans are mentioned but not included in the figures. This would be very valuable to include, particularly relative to the unique alveolar anatomy that is described. Could you present images of computer models developed from the CT scan? It would be really nice to show the dentaries in dorsal and medial views without the tooth replicas added during reconstruction, to better verify the potentially edentulous nature of the anterior left dentary...as it is, the anatomy is a little hard to follow without some figures. Comparative measurements would also be useful.

- The C5 vertebral anomalies are hard to interpret with the photos. Are the areas in the various colors (Figure 4) showing equivalent things? If possible, please include a view of the "normal" opposite side of the vertebra for comparison. Surface models from CT scans might be a better presentation (or an additional presentation).

- For the humerus (and elsewhere), reference is made to an "abnormal" morphology. This should be accompanied by at least a brief description (and perhaps an illustration) of the 'normal" morphology. Here and elsewhere, additional measurements of selected structures (e.g., the trough-like marks that are described) would be beneficial.

- p. 18 - 19: Regarding ossification of interspinous ligaments (and other vertebral ligaments), Farke & O'Connor 2007 also discussed this for Majungasaurus, noting that it occurs here and also in Masiakasaurus, and O'Connor 2007 further considers the issue. Farke & Sertich (2013) discussed similar ossifications in the abelisauroid Dahalokely tokana, also. These papers should be cited here (particularly O'Connor 2007).

- p. 20: Potential tramatic pathologies in vertebrae were also described for ceratopsian and hadrosaur vertebrae by Darren Tanke and colleagues. See the reviews published in the recent Horned Dinosaur and Hadrosaur volumes from IUP.

- p. 20: "while an injuried caudal in the abelisaurid Majungasaurus (Farke & O'Connor, 2007) may indicate cannibalistic behavior." We (Pat and I) took a far more conservative approach to interpreting the tail truncation and other vertebral anomalies in Majungasaurus than implied here. Quoting p. 183 from our 2007 paper: "the types of abnormalities reported here for Majungasaurus are not amenable to precise reconstruction of a traumatic event." It may well have been cannibalism, or intraspecific fights, or whatever, but there is absolutely no way to know for certain. Maybe the animal got its tail wedged between two rocks?

- p. 24, lines 577 - 579: Could the pits be related to avulsion of ligaments or tendons? Failure of ossification during development? Healing responses are a diverse lot.

- p. 28: The more appropriate term rather than "monospecific" might be "monodominant." I am hesitant to ascribe all or even most monodominant assemblages to gregarious behavior, if only because monodominant associations are common across many organisms - e.g., turtles, cockroaches, alligators, etc. Sometimes an association reflects gregarious aggregations - sometimes it reflects assembly around a resource. Sometimes both.

CITATIONS
Farke, A. A., and J. J. W. Sertich. 2013. An abelisauroid theropod dinosaur from the Turonian of Madagascar. PLOS ONE 8:e62047.

Farke, A. A., E. D. S. Wolff, and D. H. Tanke. 2009. Evidence of combat in Triceratops. PLOS ONE 4:e4252.

O’Connor, P. M. 2007. The postcranial axial skeleton of Majungasaurus crenatissimus (Theropoda: Abelisauridae) from the Late Cretaceous of Madagascar. Society of Vertebrate Paleontology Memoir 8:127–162.

Tanke, D. H., and B. M. Rothschild. in press. Paleopathologies in Albertan ceratopsids and their behavioral significance; pp. in M. J. Ryan, B. J. Chinnery-Allgeier, and D. A. Eberth (eds.), New Perspectives on Horned Dinosaurs. Indiana University Press.

·

Basic reporting

A very thorough atlas of the pathologies of an interesting specimen. It is important to have these kind of details of a single individual summarized in a single manuscript, to better understand the lives of particular specimens.

Experimental design

The text is clear and descriptive, as are the photographs.

Validity of the findings

The diagnoses of the various pathologies are consistent with observations in extant animals.

While I happen to agree with the authors that the possibility of gregarious feeding behavior in Allosaurus is a possibility (and may go some way to explaining the disparity in size between the larger predators and larger herbivores in the Morrison community), it might also be argued that the high pathology load in multiple individuals of Allosaurus was indicative of a low metabolic rate: that is, lower nutrient demand might allow longer periods of time between feeding, so injuries which might be catastrophic for an endotherm might be sustainable in an animal that feeds only rarely. (I am not advocating this position, but do point out it might be a potential alternative.)

Comments for the author

Corrections & Comments
p. 24, line 552 USNM 4734 also has a series of factures in the left ribs which align horizontally (also the same side as the pathological dentary and scapula). However, Gilmore did not describe these, and Chure’s description of them remains unpublished.

p. 29, line 680 Tyrannosaurus rex is also known from multiple individuals in the same locality: FMNH PR 2081 was discovered with smaller individuals of Tyrannosaurus (Larson, N.L. 2008. One hundred years of Tyrannosaurus rex: the skeletons. Pp. 1-55 in P. Larson & K. Carpenter, Tyrannosaurus rex, The Tyrant King. Indiana Univ. Press).

---

## Round 0.2 · Minor Revisions

You have addressed all of the scientific points raised by the reviewers - thank you for your thoroughness in that effort. In reading through the revised manuscript I found a handful of minor points of grammar or word choice that need to be addressed. Once these are dealt with, I look forward to accepting the manuscript for publication.

Line numbers here refer to the revised Word document with changes tracked.

Line 166: "To allow the reader an own assessment of the pathologic structures described" Suggest rewording as, "To allow readers their own assessments..."

Line 212: "However, one has to keep in might" Keep in mind, rather than keep in might

Line 281: "pseudarthrosis are associated" should be plural 'pseudoarthroses'

Line 410: "It is possible that the flexor tubercle was never" Should that be the extensor tubercle instead of the flexor tubercle? I'm not following the argument otherwise.

Line 414: "phalange" The correct singular form of 'phalanges' is 'phalanx'.

Line 648: "healed trauma long time ago" Awkward. Perhaps "trauma that healed long before the animal's death"?

Line 784: "allowed the survival of injuries" Suggest replacing with, "survival of the animal despite its injuries"

Line 785: "common survival of injuries" Suggest replacing with, "survival of the injured individual"

Line 789: "oftentimes appears with" Suggest replacing with, "is frequently found with"

Lines 781-787: This new section on alternative hypotheses is a valuable addition. You might mention at the end that the hypotheses of lower metabolic rate or gregarious behavior are potentially testable through other lines of evidence. If either of those alternative hypotheses is falsified or supported by further testing, it could lead to a reinterpretation of this and other pathological individuals.

---

## Round 0.3 · accepted · Accept

Thank you for your patience and attention to detail in revising the manuscript. It looks great, and I am happy to accept it for publication in PeerJ.

You have the option to publish the peer review history alongside the paper. That decision is entirely up to you and will have no bearing on how the paper is treated. I think there is much to be gained from publishing the peer reviews - this was a good example of constructive reviews improving an already solid paper. I leave it in your hands.